https://doi.org/10.1038/s41467-019-12400-5　　**OPEN**

# Distinct transcriptional roles for Histone H3-K56 acetylation during the cell cycle in Yeast

Salih Topal[1], Pauline Vasseur[2,3], Marta Radman-Livaja [2,3] & Craig L. Peterson[1]*

Dynamic disruption and reassembly of promoter-proximal nucleosomes is a conserved hallmark of transcriptionally active chromatin. Histone H3-K56 acetylation (H3K56Ac) enhances these turnover events and promotes nucleosome assembly during S phase. Here we sequence nascent transcripts to investigate the impact of H3K56Ac on transcription throughout the yeast cell cycle. We find that H3K56Ac is a genome-wide activator of transcription. While H3K56Ac has a major impact on transcription initiation, it also appears to promote elongation and/or termination. In contrast, H3K56Ac represses promiscuous transcription that occurs immediately following replication fork passage, in this case by promoting efficient nucleosome assembly. We also detect a stepwise increase in transcription as cells transit S phase and enter G2, but this response to increased gene dosage does not require H3K56Ac. Thus, a single histone mark can exert both positive and negative impacts on transcription that are coupled to different cell cycle events.

[1] Program in Molecular Medicine, University of Massachusetts Medical School, 373 Plantation Street, Worcester, MA 01605, USA. [2] Institut de Génétique Moléculaire de Montpellier, UMR 5535 CNRS, 1919 Route de Mende, 34293 Montpellier Cedex 5, France. [3] Université de Montpellier, 163 rue Auguste Broussonnet, 34090 Montpellier, France. *email: Craig.Peterson@umassmed.edu

Early biochemical studies identified nucleosomes, the building blocks of chromatin, as physical barriers at different stages of transcription[1]. Nucleosomes are arranged in a regularly spaced manner, and nucleosome positioning is influenced by DNA sequence and regulated by chromatin remodeling enzymes[2]. While nucleosomes can be found at both genic and nongenic regions, promoters and termination regions are largely encompassed by nucleosome depleted regions (NDRs)[3]. NDRs are flanked by highly positioned nucleosomes (+1 and −1), where the +1 nucleosome either contains the transcription start site (TSS) or is located immediately upstream[2]. Interestingly these promoter-proximal nucleosomes are highly dynamic, showing rapid disassembly and reassembly through replication-independent nucleosome turnover[4]. One of the main features of these dynamic nucleosomes is the acetylation of lysine 56 on histone H3, and this modification has been shown to enhance the turnover of promoter-proximal nucleosomes, yielding a positive-feedback loop[5,6]. Due to its enrichment at promoters, several studies have shown a role of this modification in transcriptional activation of a select group of genes[7–9]. In addition, Weiner and his colleagues showed that H3K56Ac levels correlate with transcriptional activity[10]. In accordance with this role in yeast, H3K56Ac has also been shown to associate with the core transcriptional network in mammals, and has been suggested to promote pluripotency[7,8].

The acetylation of H3K56 is catalyzed by the histone acetyltransferase (HAT) Rtt109 in budding yeast[11], and unlike many other HATs, Rtt109 can only use free histones as a substrate for H3K56 acetylation[12]. Consequently, newly assembled nucleosomes are enriched for H3K56Ac, including nucleosomes behind replication forks, DNA repair foci, and promoter-proximal nucleosomes undergoing replication-independent nucleosome turnover[6,13,14]. Rtt109 does not share sequence homology with other known HATs, but the crystal structure of Rtt109 shows resemblance to the mammalian HATs, p300 and CBP[15], which catalyze H3K56Ac in mammals[16]. Two histone chaperones, Vps75 and Asf1, are required for Rtt109 acetylation activity in vitro, influencing its substrate specificity[12,17]. When associated with Asf1, the Rtt109-Asf1 complex preferentially acetylates H3K56, whereas the Vps75–Rtt109 complex prefers to acetylate H3K9[12,18,19]. Consistent with these in vitro studies, an *asf1Δ* causes larger reductions in H3K56Ac levels compared to H3K9Ac or H3K23Ac in vivo[18,19]. Several previous studies of steady state RNA levels failed to observe significant changes in transcription due to inactivation of Rtt109. We uncovered genetic interactions between Rtt109 and the nuclear exosome, suggesting that post-transcriptional events may mask the transcriptional impact of H3K56Ac loss[20].

In yeast, H3K56Ac is also incorporated during replication-coupled nucleosome assembly[13]. During replication, nucleosomes are disassembled ahead of the fork and these parental nucleosomes are segregated to the two daughter strands. Newly synthesized histones, marked with H3K56Ac, are then deposited to "fill in the gaps"[21]. H3K56Ac facilitates nucleosome deposition, as this mark enhances the binding of histone H3 to the key histone chaperones, chromatin assembly factor 1 (Caf1) and Rtt106, as well as promoting the subsequent binding of Caf1 to PCNA[13,22]. In the absence of H3K56Ac, nucleosome assembly is delayed, and this leads to increased genome instability[13]. For instance, an *rtt109Δ* mutant or strains harboring a H3K56R substitution derivative show increased DNA damage and sensitivity to genotoxic agents and replication stress[11]. Notably, persistent, genome-wide H3K56Ac also leads to genomic instability, and consequently the high, S phase levels of H3K56Ac are erased in G2/M by the Sirtuin family deacetylases, Hst3 and Hst4 (Sirt6 in mammals), whose expression peak in late S/G2 phase[23,24].

During S phase, replication doubles the gene dosage, but several studies have found that transcription does not immediately increase following gene duplication[25,26]. Indeed, early studies in both fission and budding yeast found that the mRNA synthesis rate following replication remained similar to that of G1 cells, until late S or G2 phase where a step-wise increase in mRNA synthesis eliminated this buffering phenomenon[27–29]. A similar phenomenon has also been observed in mammalian cells, and in this case the buffering is due to cell cycle dependent changes in the frequency of transcriptional bursting[30]. The buffering of transcription in response to gene dosage is thought to be a mechanism to couple the concentration of gene expression products to both DNA content and cell size. Interestingly, recent studies have implicated H3K56Ac in the buffering of mRNA synthesis for early replicating loci in yeast, though how this histone mark exerts such a repressive role is not clear[31].

Here, we utilize native elongating transcript sequencing (NET-seq)[32] and transient-transcriptome sequencing (TT-seq)[33] to dissect the role of H3K56Ac in transcription during G1, S, and G2 phases of the cell cycle. Using these methodologies, we find that H3K56Ac globally activates gene transcription throughout the yeast cell cycle, and find that it regulates the distribution of RNA Pol II throughout gene bodies, suggesting that enhanced, replication-independent nucleosome turnover stimulates transcription. In contrast, during S-phase, H3K56Ac inhibits transcription of both coding and noncoding RNAs immediately following replication fork passage, an activity that correlates with its role in promoting efficient nucleosome assembly. Indeed, we observe a similar repressive function during S phase for components of the primary nucleosome assembly machinery, Cac1 and Rtt106. Furthermore, we find that global transcription increases ~2× in G2 phase in the presence or absence of H3K56Ac, indicating that this histone mark is not essential for buffering of gene dosage during S phase.

## Results

**H3K56Ac has only a minor impact on steady-state RNA levels.** Previously, we found that inactivation of Rtt109 had little impact on steady state RNA levels, as assayed by high-density DNA tiling arrays[20]. As an alternative strategy, we isolated total RNA from asynchronous cultures of isogenic wild type (WT) and *rtt109Δ* strains, and RNA levels were analyzed by RNA-seq. Importantly, *Schizosaccharomyces pombe* cells were "spiked-in" during cell harvest to provide an internal normalization control for sequencing libraries. Consistent with previous tiling array analyses, inactivation of Rtt109 led to only minor changes in the yeast transcriptome (Supplementary Fig. 1). Using DeSeq analysis, the likelihood ratio test showed only 178 upregulated genes (≥1.5 FC, false-discovery rate (FDR) ≤ 0.05) and 148 downregulated genes (≥1.5 FC, FDR ≤ 0.05) in the *rtt109Δ* compared to WT (Supplementary Fig. 1A). GO-term analysis for downregulated or upregulated genes implicates genes in metabolic pathways (Supplementary Fig. 1C, D). Interestingly, the *HTA1/HTA2* (encoding histone H2A) and *HTB1/HTB2* (encoding histone H2B) gene pairs are upregulated 1.5- to 1.8-fold, while levels for *HHT1/HHT2* (encoding Histone H3) and *HHF1/HHF2* (encoding Histone H4) were not significantly changed. Together, these results show that loss of H3K56Ac has only a minor impact on the steady state RNA transcriptome.

**H3K56Ac globally activates nascent transcription.** To investigate the possibility that post-transcriptional events might mask a more global impact of H3K56Ac on transcription, nascent elongating transcript sequencing (Net-seq) was employed to more directly assess transcription in both WT and *rtt109Δ* strains.

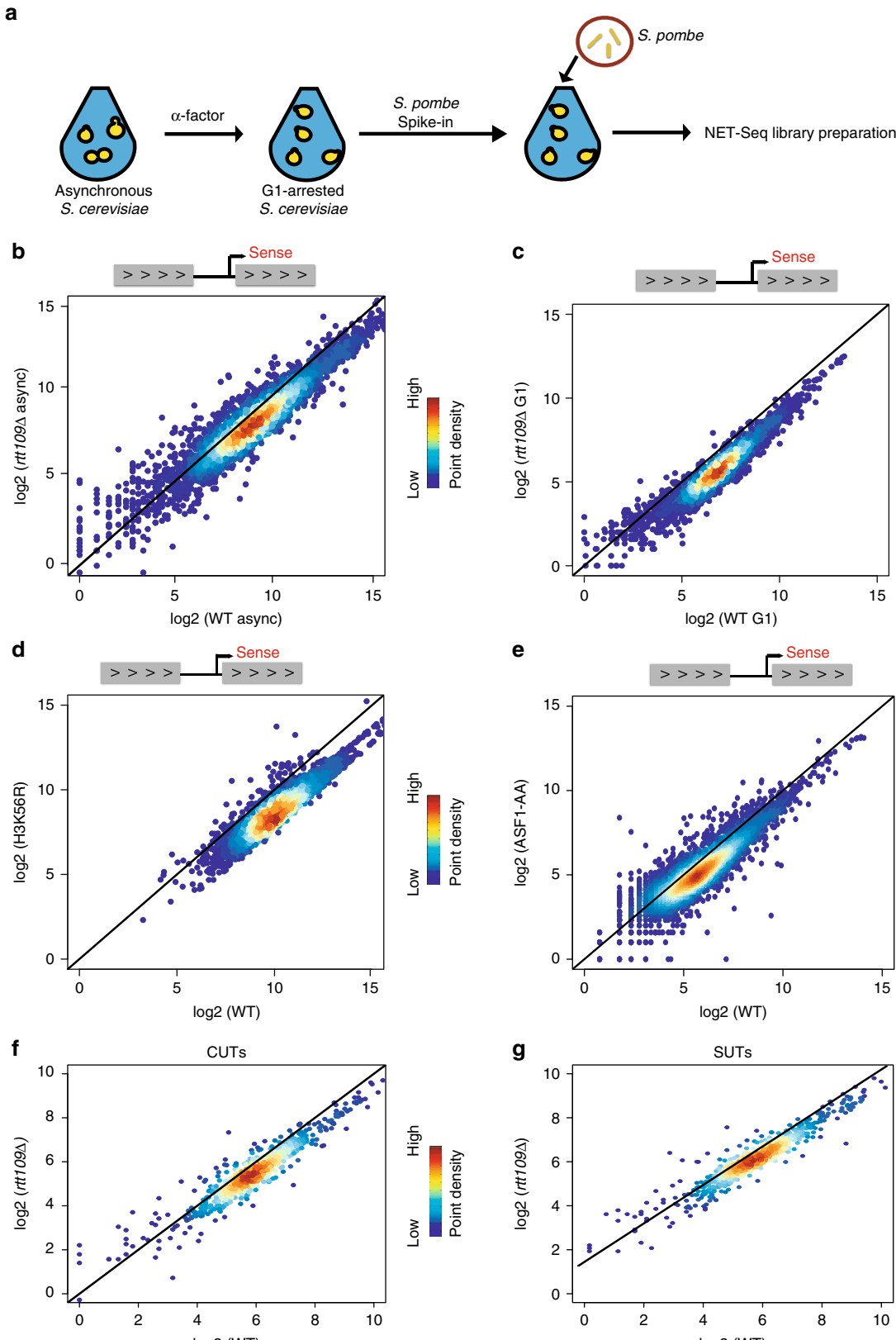

**Fig. 1** H3-K56Ac globally activates transcription (**a**). Experimental design for NET-Seq. Unless indicated, cells were arrested in G1 by alpha factor (αFT) for 1.5 h. Following the addition of control, *S. pombe* cells, nascent RNAs associated with RNAPII were isolated and sequenced. **b**–**e** NET-seq scatterplots showing log$_2$ mean intensity values for an average of two biological replicates ($n = 2$) for all coding nascent transcripts ($N = 5302$) between wild-type and mutant cells. **b** asynchronous WT and *rtt109Δ* cells; **c** G1-arrested wild-type and *rtt109Δ* cells; **d** G1-arrested WT and H3K56R cells; **e** conditional depletion of Asf1 for 1 h in G1-arrested cells (Asf1-AA), compared to WT; **f** nascent transcript levels for Cryptic Unstable RNAs (CUTs) between G1-arrested wild type and *rtt109Δ* cells; **g** nascent transcript levels for Stable Unannotated RNAs (SUTs) between G1-arrested wild-type and *rtt109Δ* cells

Similar to RNA-seq analyses, *S. pombe* cells were added during cell harvest as an external spike-in for library normalization (Fig. 1a). In contrast to the results from RNA-seq analyses, Net-seq revealed a global decrease in nascent RNA transcripts for gene coding sequences after Rtt109 inactivation (2744 genes down-regulated ≥ 1.5 FC, FDR ≤ 0.05; Fig. 1b). Strikingly, decreases in nascent transcript levels were more dramatic when RNA was analyzed from cells arrested in the G1 phase of the cell cycle as compared to asynchronous cells (Fig. 1b, c). In this case, 3299 genes decreased by >1.5-fold (FDR ≤ 0.05) (see also Supplementary Fig. 2A). Notably, Rtt109-dependent acetylation of H3-K56 is largely restricted to promoter-proximal nucleosomes in G1 cells[5,6], suggesting that replication-independent nucleosome turnover is responsible for a global, positive role in transcription.

Rtt109 acetylates H3-K9, H3-K14, and H3-K23 in addition to H3-K56[19]. To ensure that the global decreases in nascent transcription are due solely to the loss of H3K56Ac, Net-seq analyses were performed in G1-arrested cells harboring a H3K56R substitution derivative that mimics unacetylated H3-K56. In this case as well, a global decrease in nascent transcript levels was observed (Fig. 1d). One additional possibility is that long-term growth in the absence of H3K56Ac may lead to persistent changes in nucleosome assembly/positioning due to roles for H3K56Ac during S phase, and that these changes are responsible for the decreases in nascent transcription. To test this possibility, we used the anchor-way system to conditionally deplete the Asf1 subunit of the Rtt109 HAT complex from the nucleus in G1-arrested cells[34]. Strikingly, rapid depletion of Asf1 from G1-arrested cells (Asf1-AA) also showed global decreases in nascent transcript levels (Fig. 1e). Furthermore, the *rtt109Δ*, H3K56R, Asf1-AA strains all showed comparable numbers of genes with downregulated levels of nascent transcripts, with over 2000 genes decreased at least 1.5-fold in all mutants compared to WT (Supplementary Fig. 2C).

We also investigated nascent transcript levels of two groups of noncoding transcripts—cryptic unstable transcripts (CUTs) and stable unannotated transcripts (SUTs). Both CUTs and SUTs are short-RNA products transcribed by RNA Pol II, polyadenylated, and terminated by a pathway involving Nrd1–Nab3 and the TRAMP complex[35,36]. CUTs are rapidly degraded by the RNA exosome, and thus they are not detectable by conventional methodologies such as RNA-seq[37], while SUTs are more stable transcripts. Similar to coding transcription, we observed a global decrease in nascent transcription levels of both CUTs and SUTs in the absence of Rtt109 in G1-arrested cells (Fig. 1f, g; see also Supplementary Fig. 2B). Together, these results show that H3K56Ac is a global activator of transcription.

**H3K56Ac regulates RNAPII distribution within the gene body**. Analysis of genome browser views for several representative genes showed an overall decrease in nascent transcription throughout the coding regions, though it appeared that there may be greater decreases at the 5′ ends (Supplementary Fig. 2A). To investigate this further, we sorted all genes based on the size of their coding regions and plotted their nascent transcript (RNAPII) distribution (Fig. 2a). While the heatmap plot confirmed a global decrease in RNAPII levels, there was a larger decrease proximal to the TSS (Fig. 2a). The asymmetry in nascent transcript decreases was also apparent when we calculated the ratio of reads surrounding the TSS to reads surrounding the transcription termination site (TTS; Fig. 2b). In this case, the TSS/TTS ratio is decreased nearly twofold in *rtt109Δ*, H3K56R, or Asf1-AA strains compared to WT. Next, we performed metagene analyses to analyze RNAPII distributions, normalizing RNAPII occupancy levels to the individual gene expression levels and plotted these

values throughout ~5200 genes. This analysis confirms that loss of H3K56Ac leads to a change in RNAPII distribution, with a marked decrease in RNAPII distribution near the TSS (+1 nucleosome) and an increase in RNAPII levels over the gene body and near the TTS (Fig. 2c, upper panel). Similar changes in the RNAPII distribution were also observed following depletion of Asf1 or in the H3K56R strain, although the changes were less dramatic for the H3K56R strain (Fig. 2c, lower panels). This latter observation suggests that changes in RNAPII distribution along coding regions may reflect additional contributions from other histone acetylation events catalyzed by Rtt109/Asf1, such as H3K9Ac, H3K23Ac, or H3K27Ac.

The altered distribution of RNAPII towards the 3′ end of the gene body suggested that loss of H3K56Ac leads to defects in transcriptional elongation or termination. To investigate this further, we tested whether the change in RNAPII distribution was similar to that due to an altered RNAPII that has a slow elongation rate. The *rpb1-N488D* allele alters a residue that affects the polymerase catalytic center, and it slows RNAPII elongation rates in vitro and leads to defects in transcriptional elongation and termination in vivo[38,39]. Net-seq analysis of RNA isolated from an asynchronous culture of the *rpb1-N488D* strain did not show a global decrease in nascent transcript levels, with only 477 genes showing a significant change (320 downregulated genes, 157 upregulated genes ≥ 1.5 FC, FDR ≤ 0.05)(Fig. 2d). Interestingly, normalized metagene analysis for the slow RNA Pol II distribution showed a profile nearly identical to what was observed in the absence of H3K56Ac—RNAPII was decreased near the TSS, and higher RNAPII occupancy was observed over the gene body and near the TTS (Fig. 2e). Together, these results suggest that H3K56Ac promotes more efficient transcription initiation as well as transcription elongation and/or termination.

**H3K56Ac promotes transcription initiation**. Net-seq analysis monitors the level and distribution of RNAPII, but it does not directly report on the number of active RNAPII molecules. In contrast, transient transcriptome sequencing (TT-seq) monitors RNA segments synthesized during a short, 10′ pulse of 4-thiouracil[33]. In yeast, TT-seq primarily monitors the frequency of transcription initiation events due to the relatively short length of coding regions[40]. Both WT and *rtt109Δ* cells were labeled with 2.5 mM 4-thiouridine (4tU) for 10 min, and 4tU-labeled RNAs were isolated and sequenced (Fig. 3a). The results are remarkably similar to Net-seq analyses, as inactivation of Rtt109 caused a global decrease in gene coding transcription, as well as cryptic, antisense transcription (Fig. 3b–d). Omission of the sonication step of the TT-seq protocol (yielding 4sU-seq libraries) yielded similar results (Fig. 3d and Supplementary Fig. 3). These findings confirm that H3K56Ac has a global, positive impact on RNAPII-dependent transcription.

**H3K56Ac represses transcription during S phase**. A recent study implicated Rtt109 and H3K56Ac as global, negative regulators of transcription during S phase, and these investigators proposed that H3K56Ac-mediated repression functions to buffer gene-dosage imbalance following passage of the replication fork[31]. Given that we do not observe such a repressive function for H3K56Ac in G1 or asynchronous cells, we used Net-seq analysis to investigate the transcriptional impact of H3K56Ac as cells progress from G1, through S phase, and into G2. The ASF1-AA strain was arrested in G1 phase by treatment with alpha factor (αFT), and then cells were treated for 1 h with DMSO or with rapamycin to deplete the Asf1 subunit of the Rtt109 HAT complex from the nucleus. Cells were then released into a synchronous cell cycle by

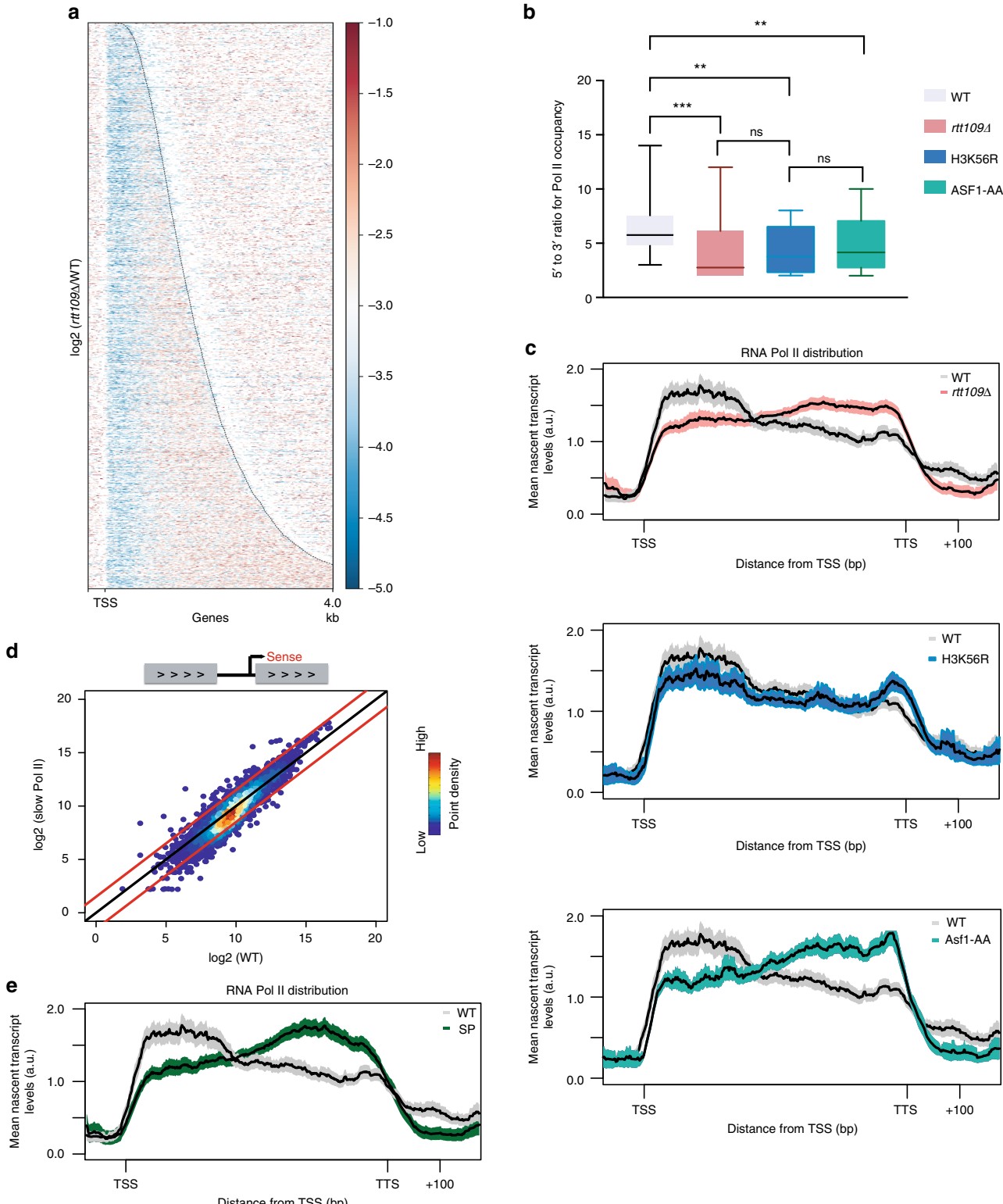

washing into media lacking αFT and containing or lacking rapamycin. Samples were collected for Net-seq analysis at time zero (G1), 30′ postrelease (early S phase), 60′ postrelease (late S phase), or 90′ postrelease (G2) (Fig. 4a). Cell cycle positions were determined by observing cell morphology, FACS analysis, and monitoring cell cycle regulated transcripts for both WT and Asf1-AA strains (Supplementary Fig. 4).

We initially analyzed nascent transcripts for all coding genes, comparing levels in WT to those in cells depleted for Asf1 (Asf1-

AA). First, we tested whether we could detect buffering of gene dosage as WT cells progressed through S phase. Consistent with buffering, WT gene expression levels in early and late S phase remained nearly identical to that of G1 cells, even though most genes had duplicated by the 60′ timepoint (Fig. 4b; Supplementary Figs. 4 and 5A–F). Interestingly, the distribution of RNAPII along coding regions was unaltered as cells transited S phase (Supplementary Fig. 5G). As cells entered G2 phase (90′ time point), the gene-expression profile showed a stepwise doubling

**Fig. 2** H3-K56Ac regulates RNA Pol II distribution within the gene body. **a** Heatmap showing $\log_2$ fold change of nascent transcript levels between wild type and *rtt109Δ* for a region beginning with the TSS and extending 4 kb downstream, ordered by gene length. **b** Boxplot showing 5′ to 3′ ratio for RNA Pol II occupancy for wild-type (gray), *rtt109Δ* (red), H3K56R (blue), and Asf1-AA (green). Reads from the TSS to 300 bp downstream were summed and divided by the sum of the reads from the TTS to 300 bp upstream. The lateral lines in the boxes represent the median, and separate upper and lower quartiles. The vertical lines represent the highest and lowest data points. Significance and *p* values were calculated using Mann–Whitney *U* test. Error bars represent standard deviation. ns is not significant. **c** Metagene plots showing RNA Pol II distributions for wild type (gray) and *rtt109Δ* (red, upper panel), H3K56R (blue, middle panel), or Asf1-AA (teal, lower panel) throughout the gene body from TSS to TTS (including 100 bp upstream of TSS and 200 bp downstream of TTS) fitted into a 500 bp window. The mean nascent transcript levels are normalized according to both spike-in numbers and each gene's individual expression level. Shaded area represents the 95% confidence interval. **d** Scatterplot showing nascent transcript levels for all coding regions between wild type and *rpb1-N488D* (slow Pol II). **e** Metagene plot showing RNA Pol II distribution throughout the gene body, as in (**c**). AU arbitrary unit

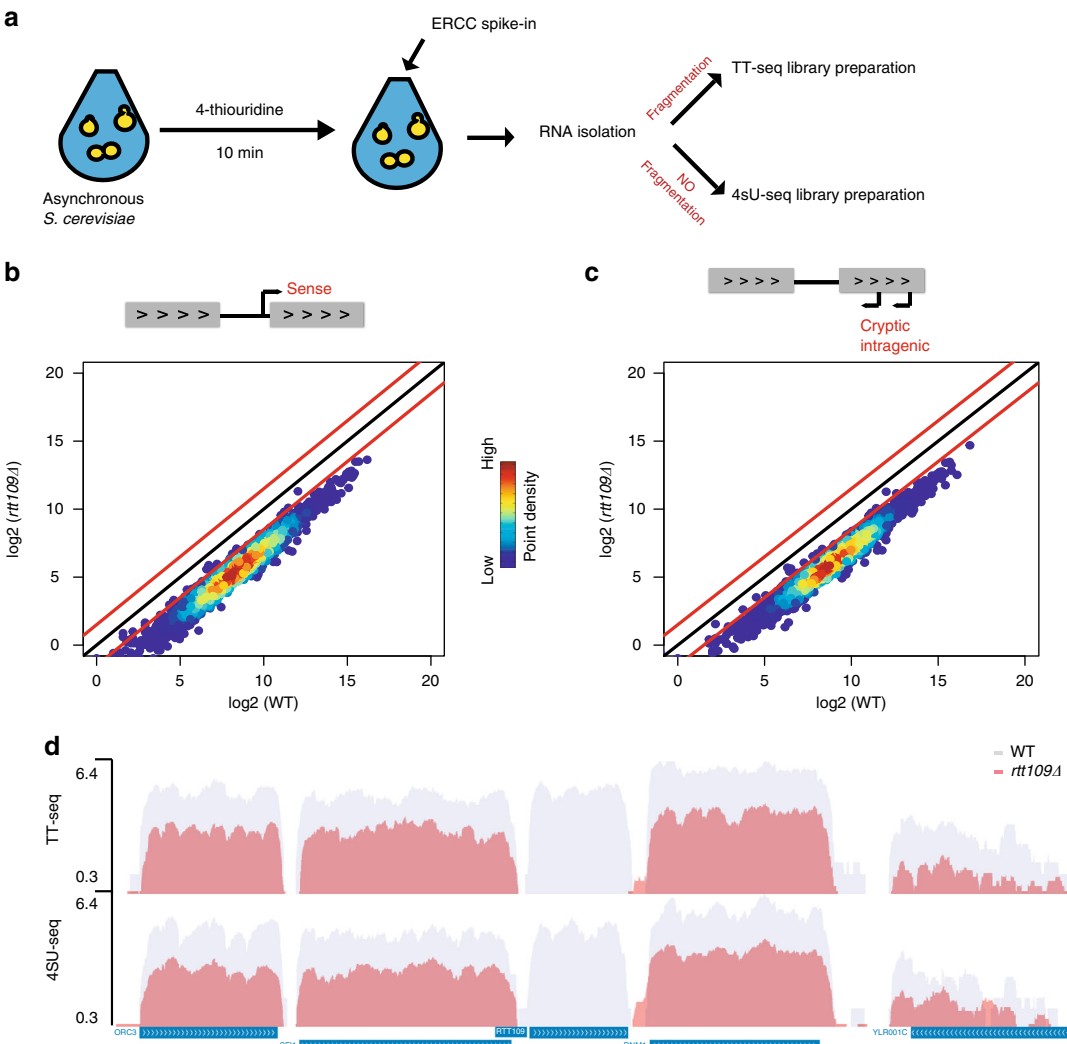

**Fig. 3** H3-K56Ac promotes transcription initiation. **a** Experimental design for TT-Seq and 4sU-Seq. Asynchronous WT and *rtt109Δ* cells were labeled with 2.5 mM 4tU for 10 min. Following fragmentation (TT-Seq libraries) or no fragmentation (4sU-Seq libraries), 4sU-labeled RNAs were isolated and sequenced. **b** Scatterplots showing coding nascent transcript levels, and **c** cryptic intragenic transcript levels between wild type and *rtt109Δ* by TT-Seq. All reads are normalized according to ERCC spike-in numbers. **d** Selected genome browser view comparing wild-type (gray) and *rtt109Δ* (red) for TT-Seq and 4sU-Seq libraries. Note the expected lack of transcripts from the *RTT109* locus from the *rtt109Δ* strain

(2.1-fold increase), compared to G1 cells, indicating a loss of dosage buffering as cells enter G2 (Fig. 4b).

As expected (see Fig. 1e), depletion of Asf1 in G1 cells led to a global decrease in nascent RNAs compared to WT cells (3085 downregulated genes ≥ 1.5 FC, FDR ≤ 0.05) (Fig. 4b), but surprisingly, the depletion of Asf1 in either early or late S phase (30′ or 60′ timepoint) did not have a significant influence on global, nascent RNA levels compared to WT (1.2-fold decrease) (Fig. 4b). However, as cells entered G2 phase, nascent transcript

levels were again significantly lower (1.7-fold decrease) compared to WT (90′ time point; Fig. 4b). Notably, as cells progressed from S to G2, transcripts increased 1.5-fold in the absence of Asf1, indicating that Asf1 may not be essential for gene-dosage buffering during S phase (Fig. 4b, Supplementary Figs. 5 and 6).

To further investigate the impact of H3K56Ac on S phase transcription, we evaluated the impact of Asf1 depletion on transcription of genes during replication fork passage. To this end, we identified 450 genes located 4 kb upstream or

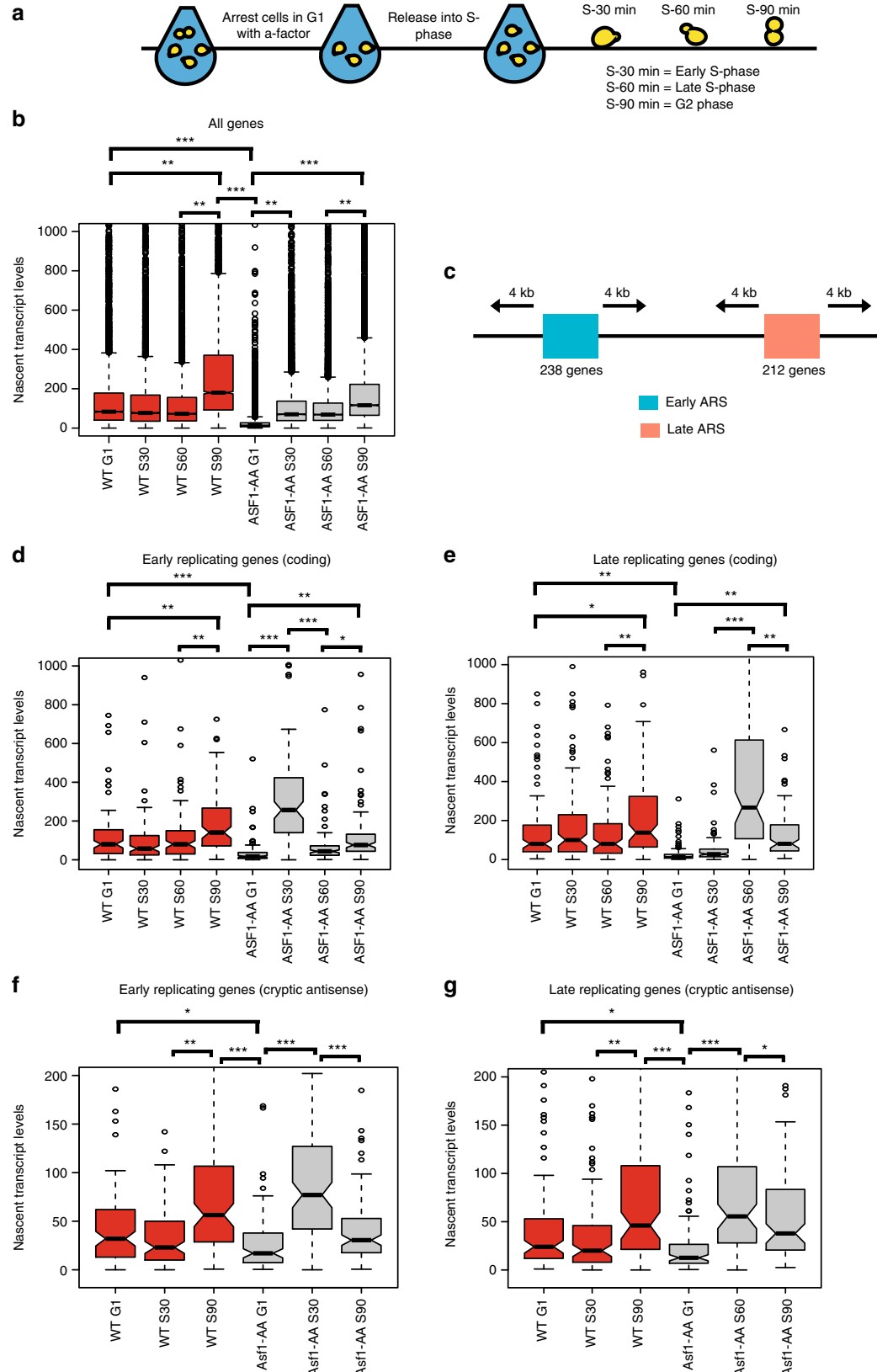

downstream of previously annotated yeast replication origins (ARSs), and then we categorized them as early- or late-replicating genes based on the known timing of ARSs[41] (Fig. 4c). In WT cells, nascent transcript levels for these genes mirrored the analysis of all genes (1.7-fold increase, comparing the G1 and G2 time points). Surprisingly, depletion of Asf1 led to a dramatic

increase (8-fold increase) in nascent transcript levels for early replicating genes in early S-phase (30′ timepoint), but expression was restored to a lower, G1 level by late S phase (60′ timepoint; Fig. 4d). Importantly, this lower level of expression in late S phase reflects productive buffering of early replicating genes, as expression of early replicating genes increased 1.7-fold as cells

**Fig. 4** Asf1 prevents promiscuous transcription during S phase. **a** Experimental design for S-phase studies. Cells were arrested in G1 by alpha factor (αFT) for 1.5 h and the histone chaperone Asf1 was depleted from the nucleus by a 1 h rapamycin treatment. Following arrest and depletion, cells were released into S-phase and time points were collected at time zero (G1), 30 min postrelease (early S-phase), 60 min postrelease (late S-phase) and 90 min postrelease (G2). **b** Boxplot showing nascent transcript levels at different time points (G1, S30, S60, and S90) between wild type (red) and Asf1-AA (gray). Significance and p-values were calculated by using Mann–Whitney *U* test. Error bars represent standard deviation. **c** Schematic showing selection of early or late-replicating genes based on previously identified early or late-replicating origins[41]. 4 kb upstream and downstream of replication origins were screened, and 238 early replicating and 212 late-replicating genes were selected for analyses. **d**–**g** Boxplots showing nascent transcript levels at different time points (G1, S30, S60, and S90) between wild type (red) and Asf1-AA (gray) for early replicating genes (**d**) or late-replicating genes (**e**) for coding regions; Boxplots showing nascent transcript levels at different time points (G1, S30, and S90) between wild type (red) and Asf1-AA (gray) for cryptic antisense transcription at the vicinity of early replicating genes (**f**) or late-replicating genes (**g**). The lateral lines in the boxes represent the median, and separate upper and lower quartiles. The vertical lines represent the highest and lowest data points. Significance and *p* values were calculated by using Mann–Whitney *U* test. Error bars represent standard deviation

entered G2 at the 90′ time point. Likewise, late-replicating genes showed a large increase (10.5-fold increase) in nascent RNAs only during late S phase (60′ time point), and these levels were decreased in G2 (90′ time point; Fig. 4e). These large increases in transcription due to Asf1 depletion were not limited to gene coding transcription, but cryptic antisense transcription showed nearly identical expression patterns (Fig. 4f, g). Similar results were also observed in synchronized *rtt109Δ* cells (Supplementary Fig. 5H, I). These results suggest that H3K56Ac transiently functions as a transcriptional repressor during the replication program.

**H3K56Ac promotes nucleosome assembly during S phase.** Previous studies have demonstrated that Asf1 and H3K56Ac stimulate nucleosome assembly in vitro and promote timely nucleosome deposition in vivo[13,42]. Thus, one possibility is that Asf1 depletion leads to partially assembled chromatin structures that create a permissive environment for transcription, and that slow maturation of chromatin eventually restores a normal transcription level. One prediction of this model is that disruption of the Cac1/Rtt106 nucleosome assembly machinery, which does not affect H3K56Ac levels[13], should also lead to promiscuous transcription during S phase. Isogenic WT and *cac1Δ rtt106Δ* double mutants were arrested in G1, released into a synchronous S phase, and time points were harvested for Net-seq analyses. Remarkably, nascent transcription in the *cac1Δ rtt106Δ* double mutant closely mirrored the patterns observed after Asf1 depletion (Fig. 5a, b). For instance, transcription of early replicating genes was dramatically increased (25-fold increase) in the *cac1Δ rtt106Δ* double mutant during early S phase (30′ time point), and levels decreased back to WT levels by G2 phase (90′ time point). Likewise, transcription of late-replicating genes was increased (13-fold increase) during late S phase and into G2 (60′ time point). Collectively, these data suggest that disruption of nucleosome assembly behind the replication fork creates a promiscuous environment for transcription.

Previous studies found that H3K56Ac had a weak impact on replication-coupled nucleosome assembly in vivo, though these studies only monitored bulk nucleosome assembly on the lagging strand[42]. To directly assess the impact of H3K56Ac on the efficiency and timing of promoter nucleosome assembly during S phase, we used nascent chromatin avidin pulldown (NChAP) to probe nucleosome deposition and spacing on newly replicated DNA strands[43]. WT or *rtt109Δ* cells were pulsed labeled with the nucleotide analog, 5-ethynyl-2′-deoxyuridine (EdU), followed by a thymidine chase. Chromatin was then digested with micrococcal nuclease (MNase), and the isolated DNA fragments subjected to a click reaction that adds biotin to the newly synthesized DNA that incorporated EdU. Biotinylated DNA was then purified with streptavidin-coated magnetic beads, and the purified DNA used for preparation of sequencing libraries.

Nucleosome profiles from all genes were aligned to their TSS, and two replicates of nascent profiles from either the WT or *rtt109Δ* strain were compared to a midlog WT standard (Fig. 6a, left panels). As observed previously, newly replicated DNA in WT cells rapidly acquired the Mnase pattern of the midlog WT control, reflecting rapid nucleosome assembly. In contrast, deposition of nucleosomes at promoters was disrupted at newly replicated DNA in the absence of Rtt109, and the disrupted pattern was largely restored by 15′ after EdU removal (Fig. 6a, right panels). Loss of Rtt109 led to a decrease in the peak-to-trough ratio of the MNase profile in nascent chromatin, consistent with poor nucleosome deposition (Fig. 6b, left panels). Furthermore, in the presence or absence of Rtt109, nucleosome spacing in newly replicated chromatin was expanded from 13 to 20 bp, but this slightly larger spacing persisted even in long-term, bulk samples from the *rtt109Δ* strain (Fig. 6b, right panel). Together, these data provide direct evidence that Rtt109, and thus H3K56Ac, is required for timely deposition of nucleosomes on newly replicated DNA, and furthermore, in the absence of H3K56Ac, the resulting disorganized chromatin structures are permissive for transcription.

## Discussion

The Rtt109/Asf1-dependent formation of H3K56Ac marks newly assembled nucleosomes throughout the yeast genome, including nucleosomes assembled during DNA repair, DNA replication, and during replication-independent nucleosome turnover[6,13,14]. Prior studies have found that hot spots for H3K56Ac are localized at promoter-proximal nucleosomes in the absence of DNA replication, reflecting their dynamic turnover[5,6]. Importantly, H3K56Ac is not simply a passive mark, as H3K56Ac enhances the rapid turnover of promoter nucleosomes, creating a positive feedback loop[5]. Here, we found that H3K56Ac stimulates transcription genome-wide, consistent with a positive, direct role for promoter-proximal nucleosome turnover in enhancing transcription initiation.

How does H3K56Ac promote nucleosome turnover? H3K56 is located near the very edge of the nucleosome where it is involved in a water-mediated contact with nucleosomal DNA[44]. Acetylation of H3K56 disrupts this contact, leading to enhanced "breathing" of the final 10 bp of DNA at each end of the nucleosome[45]. Importantly, however, H3K56Ac does not grossly destabilize nucleosomes, nor does it alter the ability of nucleosomal arrays to fold into 30 nm-like fibers[46,47]. H3K56Ac stimulates the activity of a subset of chromatin remodeling enzymes, such as RSC, and this may help to promote displacement of promoter-proximal nucleosomes[45]. In addition, we have found that H3K56Ac alters the H2A.Z deposition activity of the SWR1C remodeler, consistent with functional connections between H2A.Z and H3K56Ac[46]. Thus, the data indicate that H3K56Ac likely

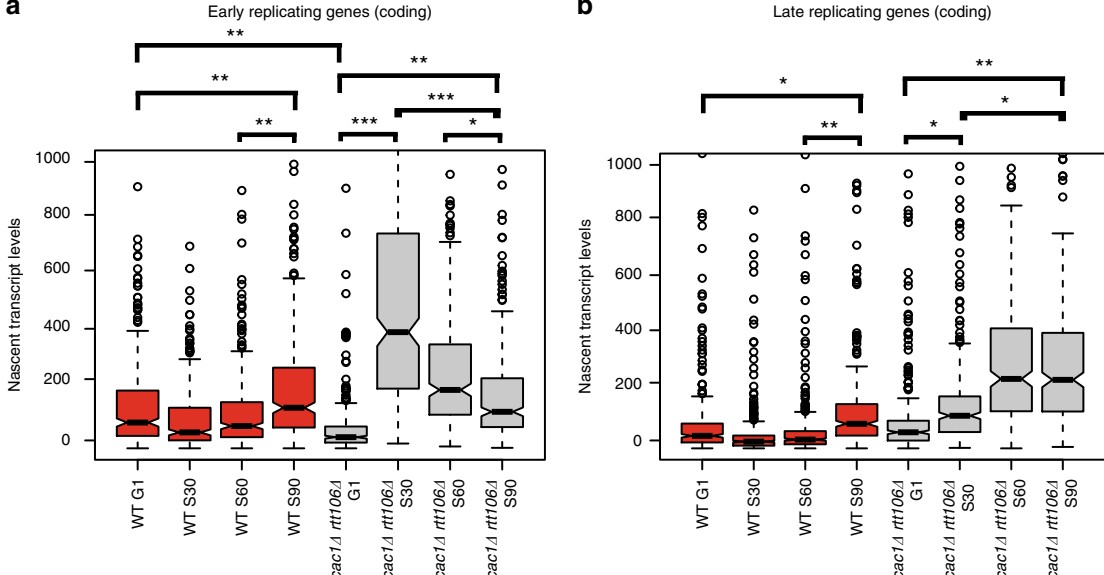

**Fig. 5** Delayed nucleosome assembly affects transcription during S phase. **a**, **b** Boxplots showing nascent transcript levels at different time points (G1, S30, S60, and S90) between wild type (red) and cac1Δ rtt106Δ (gray) for early replicating genes (**a**) or late-replicating genes (**b**) for coding regions. The lateral lines in the boxes represent the median, and separate upper and lower quartiles. The vertical lines represent the highest and lowest data points. Significance and p values were calculated by using Mann–Whitney U test. Error bars represent standard deviation

enhances promoter nucleosome turnover by altering the substrate for multiple ATP-dependent remodeling enzymes.

Our data support the idea that H3K56Ac-stimulated nucleosome turnover provides a permissive environment for transcription, globally stimulating transcription initiation at both coding and noncoding loci. We also observed that loss of H3K56Ac causes redistribution of the remaining RNA Pol II (Fig. 2), with more RNAPII located over gene bodies and near the TTS, as compared to WT cells. Indeed, loss of H3K56Ac mimics an altered RNAPII distribution due to an elongation defective, slow RNAPII. Why would loss of H3K56Ac cause polymerases to appear to slow? One attractive possibility is that the enhanced dynamics of H3K56Ac nucleosomes that are located within the coding region may promote the ability of RNAPII to elongate through nucleosomes. Consistent with a role in transcriptional elongation, both RTT109 and ASF1 show negative genetic interactions with the gene encoding the TFIIS elongation factor, Dst1[18,48]. Finally, H3K56Ac is often enriched near the 3′ end of genes, so it is also possible that loss of H3K56Ac leads to more stable nucleosomes near the TTS. Such stable nucleosomes may promote RNAPII stalling, but not release, leading to an accumulation of RNAPII at the 3′ end of coding regions.

During S phase, DNA replication provides a window of time where nucleosomes are displaced from gene promoters, and precise nucleosome positions and epigenetic marks must be restored onto newly assembled nucleosomes[21,49]. In yeast, promoter nucleosome architecture is reassembled within minutes of fork passage, whereas in Drosophila cells, nucleosomes are rapidly deposited, but reformation of gene-specific nucleosome positions is delayed for at least an hour. It has been proposed that rapidly assembled, disorganized nucleosomes may enhance transcription factor specificity[50]. Here, we have described another role for efficient and rapid nucleosome assembly behind the replication fork in which newly assembled nucleosomes repress aberrant levels of coding and noncoding transcripts. When assembly is delayed by loss of either Asf1, Rtt109, or Cac1/Rtt106, transcription is dramatically increased during replication, with levels returning to near WT levels as nucleosome assembly and positioning are restored. We note

that this aberrant transcription is higher in the absence of Cac1/ Rtt106, compared to the lack of Asf1/Rtt109, and that the changes persist into late S or G2. These differences likely reflect the greater impact of these chaperones on nucleosome assembly, as well as slower kinetics for restoration of a normal assembly pattern. In addition, we see evidence for aberrant initiation events, with increased levels of Net-seq reads upstream of coding regions or on the opposite strand (Supplementary Fig. 5J). Thus, these data provide further evidence that nucleosome assembly not only regulates transcription initiation but that it may also strengthen initiation fidelity.

Whereas depletion of Asf1 leads to a global decrease in transcription during G1 or G2 cells, there was surprisingly little impact at early or late S phase time points, with the exception of newly replicated loci. Previous work has indicated that promoter nucleosomes retain their rapid, dynamic behavior throughout S phase, as reflected by their continued enrichment for H3K56Ac[5,51]. Consequently, one might have envisioned that loss of H3K56Ac would lead to global decreases in transcription irrespective of cell cycle position. One possibility is that the positive impact of H3K56Ac may be masked in S phase by the high level of aberrant transcription that occurs due to poor nucleosome assembly in the absence of Asf1. In this case, although the majority of cells appear to move synchronously through S phase, there may be heterogeneity in the cell population whereby genes are replicating either too early or too late in a fraction of cells. Such imprecise synchrony would also lead to heterogeneity in the transcriptional consequences for loss of H3K56Ac, masking the positive role for this mark. Consistent with this view, loss of the Cac1 and Rtt106 chaperones also leads to a global increase in transcription, even in early S phase, including significant increases for late-replicating genes (Fig. 4g and Supplementary Fig. 6F).

In eukaryotic cells, mRNA synthesis rates are scaled to both DNA content and cell size. During S phase, chromosome number duplicates, but mRNA synthesis rates do not show a corresponding increase until cell size increases in G2. Currently, it is unclear if only one of the two gene copies is actively transcribed, or if both copies are only expressed at 50% of the normal level.

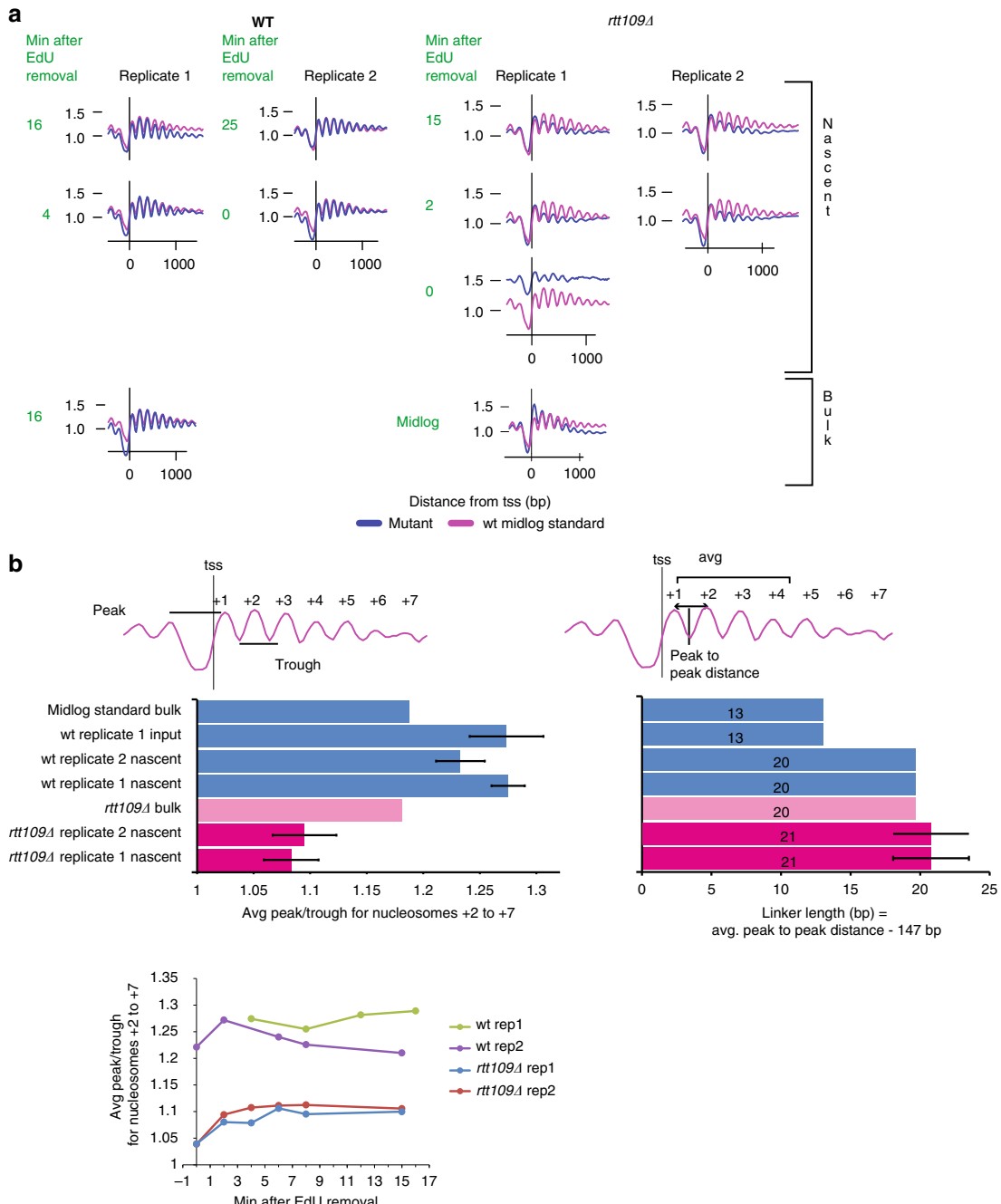

**Fig. 6** Maturation of nucleosome positioning is delayed in *rtt109Δ* cells. **a** Average TSS-aligned nucleosome profiles for all yeast genes for both WT (left panels) and *rtt109Δ* cells (right panels) from nascent (top) and bulk chromatin (bottom). Wild-type profiles are taken from[43]. Profiles for the earliest and latest time point after 5 min EdU pulse and 5 min thymidine chase (green) are shown for nascent profiles. The last time point from the corresponding total chromatin input fraction is shown for wild-type replicate 1. Wild type midlog standard profile (pink) is taken from ref. [65]. **b** Average peak/trough ratios (for nucleosomes +2 to +7) (left) and average linker length (values in the center of the bar) (between nucleosomes +1 and +2, +2 and +3, +3 and +4) (right). The error bars represent the standard deviation between time points in the EdU pulse-chase experiment. The plot on the bottom left shows the changes in the average peak/trough ratio (nucleosomes +2 to +7) after EdU removal for 2 biological replicates of wild type and *rtt109Δ*

Recently, Voichek et al.[31] devised a screen for yeast mutants that might be defective for this transcriptional buffering phenomenon by screening for mutants in which gene expression levels negatively correlated with gene replication timing[52]. They found that inactivation of either Rtt109 or Asf1 increased expression of early replicating genes, relative to late-replicating genes, as cells transitioned through S phase[31]. They concluded that H3K56Ac is essential for buffering mRNA synthesis in response to changes in gene dosage[31]. Our data also support a transient, repressive role

for H3K56Ac during S phase, but this function involves nucleosome assembly and appears to be independent of transcriptional buffering. Importantly, nascent transcripts undergo a stepwise, ~2× increase in expression as cells exit S phase and enter G2, even in the absence of Asf1 or Rtt109. Thus, nucleosome assembly is essential to maintain proper levels of transcription during S phase, but this H3K56Ac-dependent process is independent of transcriptional homeostasis mechanisms that correct for gene dosage.

Together, our findings indicate that a single histone modification, H3K56Ac, functions as a global activator of transcription, likely by enhancing promoter nucleosome turnover. In addition, H3K56Ac ensures rapid and efficient nucleosome assembly during S phase that prevents spurious transcription and reinforces transcriptional fidelity.

## Methods

**Strains**. Strains listed below were derivatives of either W303 (MATa his3-11, 15 leu2-3,112 trp1Δ ura3-1 ade2-1 can1-100) or BY4741 (MATa his3Δ1 leu2Δ0 met15Δ0). The following strains for NET-seq were modified with a 3× FLAG tag on Rpb3: rtt109Δ (this study), Asf1-AA (this study), cac1Δ rtt106Δ (this study), H3K56R (courtesy of Dr. Paul D. Kaufman, UMMS), rbp1-N488D (slow RNAPII) (courtesy of Dr. Stephen Buratowski, Harvard University). Unless otherwise noted, cells were cultivated in YPD (10% yeast extract, 20% bacterial peptone, and 2% glucose) at 30 °C. For S. pombe strain (JY741, WT Flag-Rbp3) (courtesy of Dr. Makoto Kimura, Kyushu University), cells were cultivated in YES (yeast extract, 10× aa supplement and 3% glucose). For α factor arrest, cells were grown to a density of $1.5-2.5 \times 10^7$ cells/ml in YPD and arrested by 5 µg/ml αF for 1.5 h. Arrest was confirmed by microscopic observation after 90 min.

**RNA isolation**. Cells were grown to midlog phase, and centrifuged at 1500 g for 3 min at 4 C. Following resuspension in 1 ml cold water and brief centrifugation at 4 °C, cells were resuspended in 400 µl TES (10 mM Tris-Cl pH 7.5, 10 mM EDTA, 0.5% sodium dodecyl sulphate). A 400 µl acid phenol (preheated to 65 °C) was added and samples were incubated at 65 °C for 60 min with brief vortexing. After centrifugation, top (aqueous) phase was transferred to a new tube, extracted once with chloroform, and the supernatant subjected to standard ethanol precipitation.

**RNA-seq**. Total RNA was isolated with hot acidic phenol as described above. Library preparation was performed as described[53] with the addition of an ERCC spike-in mix for two biological replicates. Briefly, total RNA was fragmented and reverse transcribed with random hexamers. Following dUTP incorporation, cDNAs were end-repaired with a mix of end-repair enzymes (T4 DNA Polymerase (3 U/µl, NEB), Klenow DNA Polymerase (5 U/µl, NEB), and T4 PNK (10 U/µl, NEB)) and A-tailing was performed. Adapter ligation was done using T4 DNA ligase (600 U/µl, Enzymatics, Inc.). After UDG treatment, final PCR was performed with indexing primers. Size of the library was determined by Fragment Analyzer and the concentrations were determined by Qubit 4.0 fluorometer (Invitrogen). Sequencing of all samples was carried out on an Illumina NextSeq 500 with a read length of 150 (paired-end).

**RNA-seq data preprocessing and normalization**. FASTQ files from paired-end libraries were collapsed by barcode and the Illumina adapter sequence was trimmed from the 3′ end. Files were uploaded and analyzed using the Galaxy web platform[54]. Reads were first aligned using Bowtie2[55,56] to Saccharomyces cerevisiae rRNA, tRNA, and RDN sequences to remove contaminating reads. Reads were then aligned to a combined version of the S. cerevisiae genome (SacCer3, SGD) and a list of ERCC spike-ins with TopHat2[57] allowing up to two mismatches. The reads were separated by their respective genomes with SAMtools[58], and only uniquely mapped reads were used for further analyses. For visualization in USCS genome browser, libraries were normalized by ERCC spike-in numbers. To compare RNA expression between samples, HTseq 0.9.1[59] was used to count the number of reads that aligned to each annotated gene. The annotation file for the S. cerevisiae genome was generated from the Xu et al. dataset[35]. Differential expression analysis was performed using edgeR[60] or in excel using the R q value package.

**NET-seq**. NET-seq conditions, immunoprecipitations, isolation of nascent RNA, and library construction were carried out as previously described[32] for two biological replicates for WT and two biological replicates for mutants (rtt109Δ, Asf1-AA, H3K56R, cac1Δ rtt106Δ, or slow Pol II) with several modifications including addition of S. pombe cells as spike-in control. Briefly, overnight cultures from single yeast colonies were diluted to an $OD_{600} = 0.05$ in 1 L of YPD. Cells were grown at 30 C until $OD_{600} = 0.8$. Rapamycin was added at a final concentration of 8 µg/mL at $OD_{600} = 0.25$ for cells with anchor-away background and cells were grown for 3 h ($OD_{600} = 0.7-0.8$). To normalize the sequencing libraries, S. pombe cells were mixed with S. cerevisiae cells at a 1:10 ratio, and the cells were harvested by filtration and cryogenic lysis. 3× FLAG-tagged RNA Pol II was immunoprecipitated and nascent RNAs were purified using miRNAeasy mini kit (Qiagen). Following ligation of pre-adenylated DNA linker onto purified nascent RNAs, RNAs were fragmented and reverse transcribed. Resulting cDNAs were circularized using DNA Circligase (Lucigen). Final PCR was performed to obtain double stranded product to sequence. Size of the library was determined by Fragment Analyzer and the concentrations were determined by Qubit 4.0 fluorometer (Invitrogen). 3′ end sequencing of all samples was carried out on an Illumina NextSeq 500 with a read length of 75 (single end).

**NET-seq data preprocessing and normalization**. NET-seq reads were processed and aligned as follows using the Galaxy web platform[54]. The adapter sequence was (ATCTCGTATGCCGTCTTCTGCTTG) removed and the random hexamer sequence was removed from the 5′ end. The 3′ ends of the reads were then trimmed for quality using FASTQ Quality Timmer by sliding window[61] with a window size of 10 and a step size of 5. The reads were trimmed until the aggregate score was ≥21. Reads were first aligned using Bowtie2[55,56] to a combined FASTA file of S. cerevisiae and S. pombe rRNA, tRNA, and RDN sequences to remove contaminating reads. Reads were then aligned to a combined version of the S. cerevisiae genome (SacCer3, SGD) and the S. pombe genome (ASM294v.2, Pom-Base) with TopHat2[57], allowing up to three mismatches. The reads were separated by their respective genomes with SAMtools[58], and only uniquely mapped reads were used for further analyses. Libraries were normalized by scaling the uniquely mapped S. pombe reads to 100,000 reads. This scaling factor was then used to scale the uniquely mapped S. cerevisiae reads. To account for differences between sequencing run depth for various NextSeq runs, the pombe-scaled WT S. cerevisiae read counts were then scaled to 1 M reads, and this additional scaling factor was included to scale the sample reads. Finally, only the 5′ end of the sequencing read, which corresponds to the 3′ end of the nascent RNA was recorded and used for downstream analyses. TSS and TTS annotation was obtained from ref. [35]. Read counts for genes and noncoding regions were obtained by summing normalized base pair reads over the region of interest. For average profiles, BAM files of biological replicates were merged and processed as above, and only genes longer than 500 bp were analyzed. Genes were scaled to 500 bp, and samples were scored in 1 bp bins using the deepTools program[62]. Reads were analyzed as in ref. [63]. To calculate 5′ to 3′ ratios, the sum of reads from 1 to 250 bp from the TSS were divided by the sum of reads 250 bp upstream of the TTS to the TTS.

**TT-seq**. TT-seq experiment was performed as described[33] for two biological replicates. Briefly, $1.5 \times 10^7$ S. cerevisiae cells were labeled with 2.5 mM of 4-thiouracil (4sU) (Sigma-Aldrich) for 10 min. Cells were harvested by centrifugation at 3000g for 2 min. Total RNA was extracted with hot acid phenol as described above. RNAs were sonicated to generate fragments of <1.5 kbp using Bioruptor Standard (Diagenode). 4sU-labeled RNA was purified from 150 µg total fragmented RNA. Labeled RNA was separated with streptavidin beads (Thermo Fisher Scientific). Strand-specific library preparation for labeled RNA was performed as described[53] and as in RNA-seq section above. After library construction, size of the library was determined by fragment analyzer and the concentrations were determined by Qubit 4.0 fluorometer (Invitrogen). Sequencing of all samples was carried out on an Illumina NextSeq 500 with a read length of 150 (paired-end).

**TT-seq data preprocessing and normalization**. TT-seq data preprocessing was essentially done as described[33]. Briefly, FASTQ files from paired-end libraries were collapsed by barcode and the Illumina adapter sequence was trimmed from the 3′ end. Files were uploaded and analyzed using the Galaxy web platform[54]. Reads were first aligned using Bowtie2[55,56] to S. cerevisiae rRNA, tRNA, and RDN sequences to remove contaminating reads. Reads were then aligned to a combined version of the S. cerevisiae genome (SacCer3, SGD) and a list of ERCC spike-ins with TopHat2[57] allowing up to two mismatches. The reads were separated by their respective genomes with SAMtools[58], and only uniquely mapped reads were used for further analyses. For visualization in USCS genome browser, libraries were normalized by ERCC spike-in numbers. To compare RNA expression between samples, HTseq 0.9.1[59] was used to count the number of reads that aligned to each annotated gene. The annotation file for the S. cerevisiae genome was generated from the Xu et al. dataset[35]. Differential expression analysis was performed using edgeR[60] or in excel using the R q value package. All sequencing data has been submitted to GEO under the accession number: GSE125843.

**EdU-thymidine pulse-chase**. EdU-thymidine pulse-chase experiment was done as described[43] for two biological replicates. Briefly, cells were grown in SCD-Ura overnight at 30 °C to an $OD_{600} = 1$. Later, cell pellets were mixed with SCD-Ura + 10 µM EdU. Thymidine (5 mM final concentration) was added after 5 or 20 min incubation with EdU at 30 °C and incubated for another 5 or 10 min. Purified EdU-labeled DNA was mixed with biotin azide solution in CuBr solution. After 2 h incubation at 37 °C, DNA was precipitated with sodium acetate and ethanol.

**Flow cytometry**. Samples were collected at certain time points at an $OD_{600}$ of 0.6–0.8. After spinning down cells, they were suspended in 70% ethanol for overnight incubation at 4 °C. Following day, cells were sonicated once at setting 3 for 5 s (Sonic Dismembrator 550, Fisher Scientific). After resuspending in distilled water and then 50 mM Tris (pH 8.0), cells were incubated in RNase A (10 mg/ml) at 37 °C for 3–4 h. After resuspending in 50 mM Tris (pH 7.5), cells were incubated in Proteinase K (2 mg/ml) at 50 °C for 1 h. Then, they were resuspended in FACS buffer (200 mM Tris-HCl (pH 7.5), 200 mM NaCl, 78 mM MgCl2). Cells were incubated with 1× Sytox Green before collecting in BD FACSDiva Software and analyzing in Flowjo v10.6.0.

**Strains used in this study**. WT Rbp3-3× FLAG ((BY4741) MATa his3Δ1 leu2Δ0 ura3Δ0 met15Δ0 rbp3::RBP3-3× FLAG::NAT)[32]

*rtt109Δ* (BY4741) MATa *his3Δ1 leu2Δ0 ura3Δ0 met15Δ0* rpb3::RBP3-3xFLAG:: NAT *RTT109*/rtt109Δ::HPH) (this study)

WT Rbp3-3× FLAG AA strain (HHY221) MAT*a tor1-1 fpr1::loxP-LEU2-loxP RPL13A-2 × FKBP12:loxP BAR1Δ::HISG* rbp3::RBP3-3xFLAG::NAT) (this study)

Asf1-AA (HHY221) MATa *tor1-1 fpr1::loxP-LEU2-loxP RPL13A-2 × FKBP12: loxP BAR1Δ::HISG* rbp3::RBP3-3× FLAG::NAT Asf1-FRB:HPH) (this study)

*cac1Δ rtt106Δ* (BY4741) MATa *his3Δ1 leu2Δ0 ura3Δ0 met15Δ0* rpb3::RBP3-3× FLAG::NAT *cac1Δ::HPH rtt106::KanMX*) (this study)

Rbp1-N488D RBP3-3XFLAG (MATa *ura3-1 leu2Δ trp1Δ::hisG his3Δ lys2Δ met15Δ rbp1-N488D* rbp3::RBP3-3× FLAG::NAT) (this study (see also Rbp1-N488D strain[38]))

*rtt109Δ-RZ17* (MATa *ade2-1 his3-11,15 leu2-3,112 trp1-1 ura3-1 can1-100 GAL psi + RAD5 + URA3::GDP-TK(7×) AUR1c::ADH-hENT1 Δrtt109::KanR*) (this study)

H3K56R Rbp3-3xFLAG (MATa *Δ(hht1-hhf1) Δ(hht2-hhf2) leu2-3,112 ura3-62 trp1 his3 + p(HHT2-HHF2, TRP1 CEN)* rbp3::RBP3-3× FLAG::NAT) (this study (H3K56R strain is courtesy of Paul Kaufman))

WT Rbp3-3× FLAG *S. pombe* strain (*h-Flag-rbp3 ade6-M216 ura4-D18 leu1*) (NBRP[64])

**Primers used in this study**. DNA linker[32]: /5rApp/(N1:25252525)(N1)(N1)(N1) (N1)(N1) CTGTAGGCACCATCAAT/3ddC/

RT primer oLSC007[32]: 5 phos/atctcgtatgccgtcttctgcttg/iSp18/cactca/iS p18/ tccgacgatcattgatggtgcctacag 3

Reverse primer oNTI231[32]: 5′ CAAGCAGAAGACGGCATACGA 3′

Custom primer for NET-seq oLSC006[32]: 5′-TCCGACGATCATTGATGGTGCCTACAG 3′

Internal RNA control for NET-seq oGAB11[32]: 5′ agu cac uua gcg aug uac acu gac ugu g 3′

**Reporting summary**. Further information on research design is available in the Nature Research Reporting Summary linked to this article.

## Data availability

All sequencing datasets are available at GEO under the accession numbers: GSE125843, GSE74090, and GSE126686. All other relevant data supporting the key findings of this study are available within the article and its Supplementary Information files or from the corresponding author upon reasonable request. The source data underlying Fig. 4d–e and Supplementary Fig. 4b–e are provided as Source Data files 1–3. A reporting summary for this Article is available as a Supplementary Information file.

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

## Acknowledgements

We thank members of the Peterson lab for helpful suggestions, and specifically Jessica Feldman (UMMS) for help with the bioinformatics analyses. We also thank the labs of Stirling Churchman (Harvard Medical School), Patrick Cramer (Munich), and William Theurkauf (UMMS) for sharing sequencing protocols and for help with troubleshooting. We thank Rahima Ziane (University of Montpellier) for constructing the *rtt109Δ* strain suitable for EdU labeling. This work was supported by grants from the NIH (GM049650 and GM122519) to C.L.P.

## Author contributions

S.T. performed all of the RNA analyses and P.V. performed the nucleosome mapping studies shown in Fig. 6. Data were analyzed by S.T., P.V., M. R-L., and C.L.P., and S.T. and C.L.P. prepared the paper.

## Competing interests

The authors declare no competing interests.
