## [Peer Review File · Nature Communications]

Reviewers' comments:

Reviewer #1 (Remarks to the Author):

In this manuscript Topal et al identify novel roles for histone H3 K56 acetylation (H3K56Ac) in transcription during the cell cycle. They report that H3K56Ac globally activates transcription and regulates the distribution of Pol II on gene bodies. They also suggest that H3K56Ac prevents promiscuous transcription in S phase and demonstrate that this histone modification is not essential for buffering of gene dosage during S phase. These are significant findings that will be of wide spread interest. The manuscript is well written, easy to follow and the data are convincing. There is only one issue that needs to be addressed. For the most part the authors analyse a combination of mutants (rtt109, H3K56R and ASF1-AA) in order to reach their conclusions about the function of H3K56Ac. The exception is the data suggesting that K56Ac prevents promiscuous transcription during S phase. In these experiments the authors only analyse the impact of Asf1 depletion. Therefore they cannot conclude that it is specifically H3K56Ac that is preventing cryptic transcription because of the multiple functions of Asf1 (e.g. Hir, CAF-1 and FACT pathways). In order to substantiate their claim they should repeat the experiment described in Fig 4 with the H3K56R strain.

Minor issue

“Furthermore, the rtt109D, H3K56R, ASF1-AA strains all showed comparable numbers of genes with down-regulated levels of nascent transcripts, with 2324 genes decreased at least 1.5-fold in all mutants compared to wild type (Fig. S2C).”

This sentence may require rewriting. It currently reads as if exactly 2324 genes were decreased in all three of the mutant backgrounds.

Reviewer #2 (Remarks to the Author):

Topal et al. characterize how loss of acetylation of histone H3 K56 influences transcription across the budding yeast genome. The authors probe H3 K56 function by using an acetylase mutant rtt109□, an inducible acetylase mutant ASF1-AA and a mutant histone: H3K56R which cannot be acetylated. Rather than analyzing mRNA – which is an imprecise readout of ongoing transcription – the authors measure transcription with two sensitive assays: Nascent Elongating Transcripts Sequencing (Net-

seq) and Transient Transcriptome sequencing (TT-seq). With these assays, they uncover a general, genome-wide defect in transcription initiation and elongation on G1 synchronized cells. This finding is important and was only uncovered because Topal et al measure nascent/ongoing transcription, rather than mRNA at steady state.

The authors then investigate the nature of H3 K56 acetylation during S phase. Previously Voichek et al, Science 2016 noted that loss of K56 acetylation during S phase lead to an increase in transcription; they proposed a model in which K56 acetylation would “buffer” the transcription of genes as their copy number doubles during DNA replication. Voichek et al showed an increase in transcription of newly replicated genes in *rtt109* and *asf1* mutant cells leading them to conclude that H3K56 acetylation is responsible for an active transcriptional repression i.e. “buffering”. In agreement with Voichek et al, Topal et al. show that loss H3K56 acetylation leads to transcriptional de-repression in S phase, but importantly, they find that effect is transient: it occurs after replication fork passage and the effect is lost soon after. This observation lead Topal et al. to the attractive hypothesis that the defect in transcription in S phase in K56 mutants is a result of a chromatin reassembly defect rather than to an active transcriptional repression mechanism (buffering). Indeed, H3K56 acetylation deficiency has been shown to result in defective chromatin reassembly in the wake of the replication fork passage (Yadav et al). Consistent with their hypothesis Topal et al show a similar, transient alteration in transcription when other chromatin assembly reassembly factors are mutated

Altogether, this manuscript is important as it helps clarify the role of H3K56 acetylation in transcriptional control. The authors provide good evidence for a positive transcriptional role for the acetylation of H3K56 in G1 and; most significantly, they provide an attractive explanation for the loss of transcription control observed in S phase in H3K56 acetylation mutants. This latter finding will be of significant interest to many and will likely dispel the idea that H3K56 acetylation buffers copy number changes in S phase. The manuscript is limited in scope and could be improved with more data: a demonstration the S phase isn't altered in K56 acetylation mutants, that changes in transcription correlate with DNA copy number; and an analysis of whether certain classes of genes (TFIID etc.) are more sensitive to loss of K56 ac in S phase and G1 would be beneficial.

Minor Points:

1. The article should edited to simplify and clarify the message of the paper.
- The title of the paper and the titles of the paragraphs should be amended explicitly convey the message of the paper.

- Figure 1 and figure 3 could be merged as they are conveying the same message
- Figure 4 and 5 could be merged
- Figure 6 is confusing and should be edited and reorganized to be more understandable to a general audience.

Reviewer #3 (Remarks to the Author):

Topal et al. investigate the role of H3K56ac in transcription throughout the cell cycle in budding yeast, primarily using NET-seq to capture nascent transcripts. They find that transcripts are largely downregulated in the absence of H3K56ac (using an *rtt109* knockout strain and a point mutation at H3K56), suggesting this mark plays a role in activating transcription. The authors find a primary role for H3K56ac in transcription initiation. In addition, they observe that H3K56ac promotes nucleosome reassembly after replication and represses non-productive transcription following replication fork passage. The approaches used here is an advance over previous studies in that nascent transcripts, rather than steady-state mRNA levels, and ongoing transcription are investigated. Furthermore, the use of synchronization to determine the role of H3K56ac throughout the cell cycle uncovers a more specific role for this mark. While these findings provide important insights into the role of H3K56ac and the data presented in the manuscript appears rigorous, there is a lack of coherence and completeness in parts of the approach and the manuscript which make it more difficult to support some of the conclusions. Specific concerns about this are outlined below:

1. While Fig 2B shows a boxplot of RNA pol II occupancy for wt, *rtt109*KO and *Asf1-AA* mutants, the metagene plot shown in Figure 2C is more useful for monitoring the RNA pol II levels across the gene. Does H3K56R show the same pattern as *rtt109*KO in this plot? This data should be available based on the experiments performed, therefore it should be added to further support the conclusion that this pattern is specific to H3K56ac.

2. The conclusions from Figure 2D-E are unclear. The authors state that the similarity in the metagene plots (Figs 2C and 2E) for the "slow" RNA pol II mutant suggests that *Rtt109*/H3K56ac also contributes to elongation/termination. However, they do not observe a similar decrease in the *rpb1* mutant in global transcription as the *rtt109* mutant, suggesting that there are differences in the defects in each mutant. In addition to the correlation between the metagene plots, can they provide other, more direct evidence that supports their conclusion?

3. Figure 4G- For the cryptic antisense transcripts in the late-replicating regions, timepoints S30 and S90 are shown. However, as shown in Figure 4E, the peak expression of the late replicating genes in the mutant is at S60 (and S30 for the early genes). Are the data in Figure 4G mislabeled? Or is there an explanation as to why the S30 timepoint only is shown for the cryptic antisense transcripts instead of S60?

4. The overall conclusion for Figure 4 is that Asf1 acts to transiently repress genes following replication. Have the authors ruled out that the observed effects are not due to changes in histone dosage in the Asf1-AA mutants during the cell cycle compared to wildtype? Also, these experiments are proposed as a means to explain the role of H3K56ac in transcription during S phase, but the rtt109 mutant is not directly tested. Do the authors have data to suggest that the rtt109 or H3K56R mutant show similar patterns?

5. Figure S4- are the gene expression plots of cell-cycle regulated genes shown just for wildtype cells? It would be helpful to show the same genes in the Asf1-AA mutant to show that the timing of the transcriptional program is similar in both strains (also informative for point 4 above in regards to the histone genes).

6. Figure 5: The authors use cac1 rtt106 double mutant strains to demonstrate that incomplete nucleosome assembly leads to a similar increase in transcription during S phase, followed by return to normal transcription levels in G2. They hypothesize that in both Asf1-AA and cac1 rtt106, there is only partial chromatin assembly and chromatin requires time to mature following replication fork passage before its becomes repressive again. Their conclusion is based on the correlation of the similar responses in both mutant strains, however there are some clear differences. In cac1 rtt106 double mutants, there is a much greater increase in transcription during S phase than in the Asf1-AA strain (for early replicating genes, 25-fold versus 8-fold). Also, for late replicating genes, there is no return to pre-replication levels during G2. If their model is correct, can the authors speculate on what might contribute to these differences? This would be a useful addition to the discussion.

7. The presentation of the data in Figure 6A is not particularly easy to follow. Why are different chase timepoints taken for the wildtype and mutant cells and for each replicate? For 6B, are the peak/trough measurements for a single timepoint or across timepoints? It seems like across timepoints would be the most uniform. Why only 1 replicate of wildtype in the timecourse graph on the bottom left? While the conclusions drawn from this figure are logical, the data are not presented in a straightforward manner, making it difficult to interpret.

Minor comment: Figure 4F and G have been mislabeled as Figure 5 in the manuscript text.

We thank all three reviewers for their thoughtful and insightful comments on our manuscript. We have added new bioinformatics analyses to the paper as detailed below, as well as changes to the text. Below are responses to each reviewer comment:

REVIEWER 1

We thank this reviewer for their positive enthusiasm for our work. The reviewer had one major and one minor comment:

In these experiments the authors only analyse the impact of Asf1 depletion. Therefore they cannot conclude that it is specifically H3K56Ac that is preventing cryptic transcription because of the multiple functions of Asf1 (e.g. Hir, CAF-1 and FACT pathways). In order to substantiate their claim they should repeat the experiment described in Fig 4 with the H3K56R strain.

We attempted to perform our S phase analysis with isogenic WT and H3-K56R strains, as suggested by the reviewer. This strain background contains deletions of the histone gene clusters and have histone genes on an episome. Unfortunately, these strains grow quite slowly (especially H3-K56R), and cells released slowly from the alpha-factor block and did not show good synchrony.

Consequently, we performed the S phase analysis in an *rtt109* deletion strain. This experiment was also suggested by Reviewer 3. Now shown in a new Supplementary Figure 5H-I, we show that inactivation of Rtt109 yielded nearly equivalent results to that of Asf1 depletion – genes that replicate early in S phase show a large increase in transcription in the *rtt109*Δ strain at 30', but this increase declines to below WT levels at 60'. Subsequently, expression increases ~2x as cells enter G2 phase at 90'. These new results are fully consistent with our previous conclusion that H3-K56Ac is not involved in transcriptional “buffering” during S phase, but rather that this histone mark ensures efficient assembly of nucleosomes following replication which helps to maintain proper levels of transcription.

Minor issue

*“Furthermore, the *rtt109*Δ, H3K56R, ASF1-AA strains all showed comparable numbers of genes with down-regulated levels of nascent transcripts, with 2324 genes decreased at least 1.5-fold in all mutants compared to wild type (Fig. S2C).”*

This sentence may require rewriting. It currently reads as if exactly 2324 genes were decreased in all three of the mutant backgrounds.

This sentence has now been changed to read: “...nascent transcripts, with over 2,000 genes decreased at least...”

REVIEWER 2

We also thank this reviewer for their positive comments and noting that our “manuscript is important”. The reviewer asked that we include more data for a few points, as well as suggesting some organizational changes.

The manuscript is limited in scope and could be improved with more data: a demonstration the S phase isn't altered in K56 acetylation mutants, that changes in transcription correlate with DNA copy number;

and an analysis of whether certain classes of genes (TFIID etc.) are more sensitive to loss of K56 ac in S phase and G1 would be beneficial.

Two pieces of data are included in the manuscript showing that loss of H3-K56Ac does not alter S phase. (1) Supplementary Figure S4A shows FACS analysis of wildtype and Asf1 depletion cells as cells are released from the G1 arrest. These data show similar progression through S phase in both strains. These FACS data also support our conclusion that gene expression in WT cells remains constant throughout S phase, and that it increases globally by ~2x as cells enter G2 at the 90' timepoint. (2) We now include in a new Supplementary Figure 4B-E an analysis of cell cycle gene expression in both WT and Asf1 depletion cells. We analyzed genes encoding histones, G1/S cyclins, and G2/M cyclin. In all cases, we see similar peaks of expression in both WT and Asf1 depletion cells, demonstrating similar S phase progression.

We have investigated whether loss of H3-K56Ac impacts classes of genes differentially, though I don't think this analysis is insightful. We looked specifically at TFIID-dependent genes, SAGA-dependent genes, and genes affected by loss of Kin28 (Kin28^{is}). Not surprising, given the global impact of H3-K56Ac (Figure 1C), we find ~90% overlap between genes that decrease significantly in the *rtt109* deletion and both TFIID and Kin28^{is} data sets. We see only ~10% overlap with SAGA-dependent genes, but we believe this is due to the fact that most of these genes are highly inducible and expressed at only low levels in the media that we have used.

Minor Points:

- 1. The article should be edited to simplify and clarify the message of the paper.*

We have gone through the text and tried to make edits where appropriate

- The title of the paper and the titles of the paragraphs should be amended to explicitly convey the message of the paper.*

We have not changed the title, as it was not clear how to convey all of the points and also keeping within the character limit. We have changed several subheadings to better fit the conclusions of each section.

- Figure 1 and figure 3 could be merged as they are conveying the same message*
- Figure 4 and 5 could be merged*

We would like to keep these figures separate, as merging will create lots of very tiny panels. We have tried to make sure that each figure has a separate main point.

- Figure 6 is confusing and should be edited and reorganized to be more understandable to a general audience.*

We agree that Figure 6 can be confusing. We have tried to modify the text to explain this more clearly.

REVIEWER 3

We thank the reviewer for their careful analysis of our manuscript and their insightful comments.

- 1. While Fig 2B shows a boxplot of RNA pol II occupancy for wt, rtt109KO and Asf1-AA mutants, the metagene plot shown in Figure 2C is more useful for monitoring the RNA pol II levels across the gene. Does H3K56R show the same pattern as rtt109KO in this plot? This data should be*

available based on the experiments performed, therefore it should be added to further support the conclusion that this pattern is specific to H3K56ac.

We have now added metagene plots for both the H3K56R and the Asf1-AA data sets in a new Figure 2C. While the Asf1-AA profile looks nearly identical to that of the *rtt109* deletion, the H3K56R data does not show as dramatic of an impact on coding region re-distribution, although the accumulation of RNAPII at the 3' end is still observed. We have now added a statement to the text that some of the changes in RNAPII distribution may be due to other sites of H3 acetylation that are catalyzed by Rtt109 (e.g. H3K9, H3K23, and H3K27).

- 2. The conclusions from Figure 2D-E are unclear. The authors state that the similarity in the metagene plots (Figs 2C and 2E) for the "slow" RNA pol II mutant suggests that Rtt109/H3K56ac also contributes to elongation/termination. However, they do not observe a similar decrease in the *rpb1* mutant in global transcription as the *rtt109* mutant, suggesting that there are differences in the defects in each mutant. In addition to the correlation between the metagene plots, can they provide other, more direct evidence that supports their conclusion?*

We apologize if our explanation in the text was not clear. We had noted that the “slow” RNAPII mutant did not show a global defect in transcription, but rather it only impacted the distribution of RNAPII. We concluded that this supported two roles for Rtt109 – promoting global transcription initiation, as well as promoting a normal distribution of transcribing RNAPII (the latter phenotype in common with slow RNAPII). We have now added to the discussion a sentence stating the known negative genetic interactions between Rtt109 and Dst1 (TFIIS). These interactions are consistent with a role in transcriptional elongation. We also mentioned in the discussion that a role for H3-K56Ac in termination is consistent with the known enrichment of this mark at the 3' end of genes.

- 3. Figure 4G- For the cryptic antisense transcripts in the late-replicating regions, timepoints S30 and S90 are shown. However, as shown in Figure 4E, the peak expression of the late replicating genes in the mutant is at S60 (and S30 for the early genes). Are the data in Figure 4G mislabeled? Or is there an explanation as to why the S30 timepoint only is shown for the cryptic antisense transcripts instead of S60?*

We apologize – yes the Figure was mis-labelled. This has been corrected.

- 4. The overall conclusion for Figure 4 is that Asf1 acts to transiently repress genes following replication. Have the authors ruled out that the observed effects are not due to changes in histone dosage in the Asf1-AA mutants during the cell cycle compared to wildtype? Also, these experiments are proposed as a means to explain the role of H3K56ac in transcription during S phase, but the *rtt109* mutant is not directly tested. Do the authors have data to suggest that the *rtt109* or H3K56R mutant show similar patterns?*

As suggested by the reviewer (see below), we have analyzed histone gene expression during Asf1 depletion, and as expected, we do see an increase in histone expression in early G1 (along with all other early replicating loci). It is not clear if these 1.5-2x increases in mRNA give rise to altered levels of histone proteins. It is also not clear if changes in histone dosage would produce the observed, transient increases in gene expression for newly replicated loci.

As described above in our comments to Reviewer 1, we have now included an S phase expression analysis in the *rtt109* deletion strain (new Supplementary Figure 5H-I). These new data are fully consistent with our previous analysis of the Asf1 depletion strain, showing transient increases in transcription as genes are replicated. These data reinforce the conclusion that H3-K56Ac is not involved in S phase “buffering” of transcription, but rather that this mark promotes efficient nucleosome assembly.

5. Figure S4- are the gene expression plots of cell-cycle regulated genes shown just for wildtype cells? It would be helpful to show the same genes in the Asf1-AA mutant to show that the timing of the transcriptional program is similar in both strains (also informative for point 4 above in regards to the histone genes).

As suggested by the reviewer, we have now included cell-cycle gene expression data for the Asf1 depletion strain. In every case, we observe similar peaks of gene expression, indicating that cell cycle timing is similar.

*5. Figure 5: The authors use *cac1 rtt106* double mutant strains to demonstrate that incomplete nucleosome assembly leads to a similar increase in transcription during S phase, followed by return to normal transcription levels in G2. They hypothesize that in both *Asf1-AA* and *cac1 rtt106*, there is only partial chromatin assembly and chromatin requires time to mature following replication fork passage before it becomes repressive again. Their conclusion is based on the correlation of the similar responses in both mutant strains, however there are some clear differences. In *cac1 rtt106* double mutants, there is a much greater increase in transcription during S phase than in the *Asf1-AA* strain (for early replicating genes, 25-fold versus 8-fold). Also, for late replicating genes, there is no return to pre-replication levels during G2. If their model is correct, can the authors speculate on what might contribute to these differences? This would be a useful addition to the discussion.*

We thank the reviewer for raising this issue. We have now added a section to the discussion where we note that the *cac1 rtt106* double mutant has a more severe defect in nucleosome assembly and a longer delay before a normal pattern is re-established (compared to the *rtt109Δ*). We believe this correlates nicely with higher levels of transcription during S phase and more persistent changes in expression.

6. The presentation of the data in Figure 6A is not particularly easy to follow. Why are different chase timepoints taken for the wildtype and mutant cells and for each replicate? For 6B, are the peak/trough measurements for a single timepoint or across timepoints? It seems like across timepoints would be the most uniform. Why only 1 replicate of wildtype in the timecourse graph on the bottom left? While the conclusions drawn from this figure are logical, the data are not presented in a straightforward manner, making it difficult to interpret.

We apologize for this complicated figure. The time points are different because not all time points yielded good libraries, so they had to be discarded. Given that these are stochastic processes in cell populations, it is really the trend over the whole time course that is reproducible and key.

For Figure 6B, it is the average over all time points and the standard deviation is the deviation between time points. We have tried to make this more clear in the figure legend.

We have now added the second WT replicate to the time course graph, as suggested.

Minor comment: Figure 4F and G have been mislabeled as Figure 5 in the manuscript text.

Corrected

REVIEWERS' COMMENTS:

Reviewer #1 (Remarks to the Author):

The authors have satisfactorily addressed the issues raised by this reviewer.

Reviewer #2 (Remarks to the Author):

The revised manuscript has addressed my concerns, I recommend publication.

Reviewer #3 (Remarks to the Author):

In this revised manuscript, Topal et al have included new bioinformatics analysis, particularly of the *rtt109* mutant cells, as well as added additional bioinformatics analyses to some individual figures and clarified a few points in the manuscript. The major criticisms I had with this manuscript have been addressed. The new *rtt109* mutant data shown in Supp Fig. 5 does help to address whether or not the phenotypes observed in the *Asf1-AA* strain are specifically dependent on H3K56ac. It is unfortunate that the H3K56R mutant strains are not amenable to this experiment (is the difficulty with obtaining synchronous cultures also observed in other plasmid-based histone mutant strains?), but the *rtt109* mutant data are sufficiently convincing. Furthermore, the authors have added more data regarding the levels of histone and cell cycle genes in the *Asf1-AA* strains, which is informative, and also addressed the differences observed between *Asf1-AA* and *cac1rtt106* mutant strains in the discussion. Furthermore, the additional metagene plots shown in Figure 2 will be informative to readers, and despite the less dramatic difference in the H3K56R mutant, the authors provide a plausible explanation for this in the text. Overall, the changes that have been made improve the manuscript and I expect this study will contribute important concepts to our understanding of the role of H3K56ac in transcription throughout the cell cycle.